# Performance Evaluation of Three Antibody Binding Assays, a Neutralizing Antibody Assay, and an Interferon-Gamma Release Assay for SARS-CoV-2 According to Vaccine Type in Vaccinated Group

**DOI:** 10.3390/diagnostics13243688

**Published:** 2023-12-18

**Authors:** Minjeong Nam, Jae Hyun Cha, Sang-Wook Kim, Sun Bean Kim, Ki-Byung Lee, You-Seung Chung, Seung Gyu Yun, Myung-Hyun Nam, Chang Kyu Lee, Yunjung Cho

**Affiliations:** 1Department of Laboratory Medicine, Korea University College of Medicine, Seoul 02841, Republic of Korea; mjnam0906@korea.ac.kr (M.N.); koryun@korea.ac.kr (S.G.Y.); yuret@korea.ac.kr (M.-H.N.); cklee5381@gmail.com (C.K.L.); 2Department of Laboratory Medicine, Korea University Anam Hospital, Seoul 02841, Republic of Korea; chajh901@korea.ac.kr (J.H.C.); kkshws@naver.com (S.-W.K.); 3Division of Infectious Diseases, Department of Internal Medicine, Korea University College of Medicine, Korea University Anam Hospital, Seoul 02841, Republic of Korea; puppybin@gmail.com (S.B.K.); anthromeda2017@gmail.com (K.-B.L.); paramentor@gmail.com (Y.-S.C.)

**Keywords:** antibody binding assay, neutralizing assay, IGRA, SARS-CoV-2, vaccine, vector vaccine, mix-and-match vaccine, mRNA vaccine

## Abstract

We evaluated the performance of SARS-CoV-2 assays in the vaccinated group using receptor-binding domain antibody assays (RBD Ab assay), neutralizing antibody assay (nAb assay), and interferon-gamma release assay (IGR assay). We also compared the performance of the SARS-CoV-2 assays based on vaccine type in a large population. We collected 1851 samples from vaccinated individuals with vector, mix-and-match (MM), and mRNA vaccines. The performance of the RBD Ab assays was assessed by SARS-CoV-2 IgG II Quant (Abbott Laboratories, Sligo, Ireland), SARS-CoV-2 IgG (Beckman Coulter, CA, USA), and anti-SARS-CoV-2 S (Roche Diagnostics GmbH, Mannheim, Germany). The nAb assay was assessed by cPass SARS-CoV-2 neutralization antibody detection kits (GenScript, NJ, USA). The IGR assay was assessed by QuantiFERON (Qiagen, Venlo, The Netherlands). Median values of the RBD Ab assays and nAb assay sequentially increased after the first and second vaccinations. RBD Ab assays and nAb assay showed very strong correlations. The median values of the RBD Ab, nAb, and IGR were higher in the mRNA vaccine group than in the vector and MM vaccine groups. The agreement and correlation among the RBD Ab assays, nAb assay, and IGR assay were higher in the mRNA vaccine group than in the vector and MM vaccine groups. We compared the performance of the RBD Ab assay, nAb assay, and IGR assay based on the vaccine types using the RBD Ab, nAb, and IGR assays. This study provides a better understanding of the assessment of humoral and cellular immune responses after vaccination.

## 1. Introduction

Severe acute respiratory syndrome coronavirus 2 (SARS-CoV-2) has spread globally, causing widespread mortality and morbidity [1]. Immunization emerges as the preeminent and secure strategy for mitigating the impact of the ongoing pandemic, demonstrating unparalleled safety and efficacy in curtailing the spread of the infectious agent and averting severe clinical outcomes. The population acquires humoral or cellular immune responses against SARS-CoV-2 through vaccination and natural infection [2,3,4]. The immune response can be assessed by several tests including the SARS-CoV-2 antibody binding assay (Ab assay), SARS-CoV-2 neutralizing antibody assay (nAb assay), and interferon-gamma release assay (IGR assay) [3,4]. 

The Ab assay measures the humoral immune responses after vaccination and natural infection and has been developed for different target antigens and assay platforms [5]. This Ab assay mainly detects antibodies (Ab) (IgM, IgG, IgA, or total Ab) against the receptor-binding domain (RBD) of the spike protein (S), partial S protein, or nucleocapsid (N) protein [6]. Many Ab assays with high sensitivity and specificity can be harmonized and validated using World Health Organization (WHO) International Standard (IS) [7]. The nAb assay also measures the humoral immune responses that defend against blocking viral entry into host cells and neutralizing their biological effect in collaboration with immune cells [8]. A commercialized nAb assay assesses potential values as a surrogate for nAb that blocks the interactions between RBD and the angiotensin-converting enzyme 2 (ACE2) receptor [9]. The SARS-CoV-2 surrogate virus neutralizing test (sVNT) (GenScript Inc., Leiden, The Netherlands), obtained the first emergency use authorization for nAb detection. It is easily performed in clinical laboratories within a few hours using a commercial assay [10]. 

Many commercial assays measuring humoral immune responses are available. However, assessing the cellular immune responses is essential for understanding long-term immunity, vaccine effectiveness, and vaccine durability as memory T cells play a key role in providing sustained protection beyond the humoral immune responses [11]. IGR assays on the enzyme-linked immunosorbent assay (ELISA) platform are commonly used to detect the cellular immune responses after cytomegalovirus or *Mycobacterium tuberculosis* infection by measuring IGR values [12,13]. The IGR assay could be applied to SARS-CoV-2 infection and was designed to function without special equipment in a short testing time [14].

Many studies have evaluated the performance of RBD antibody (Ab) assays and the nAb assay among vaccinated groups. Depending on the RBD Ab assay platform, RBD Ab assays had a sensitivity of 90.0% to 97.4% and a specificity of 97.9% to 100% [15]. Moreover, the RBD Ab assay and nAb assay were associated to varying degrees [16,17,18,19]. Unlike commercial assays measuring humoral immune responses, the conventional methods of detecting cellular immune responses are complex and have not yet been standardized, requiring highly specialized facilities [20,21,22]. Moreover, few studies have comprehensively compared the performance of RBD Ab assay, nAb assay, and IGR assay for SARS-CoV-2 in a group of vaccinated people. 

RBD Ab assays, nAb assays, and IGR assays are commercially available to measure levels of humoral and cellular immunity. Therefore, we aimed to evaluate the analytical performance of the RBD Ab assay, nAb assay, and IGR assay based on the vaccine type, vector, mix-and-match (MM), and mRNA vaccines, in a large population. 

## 2. Materials and Methods

### 2.1. Study Samples

This study was conducted from March 2021 to May 2022 at the Korea University Anam Hospital, Seoul, Republic of Korea. We included subjects aged 18 years and above, comprising vaccinated subjects without a history of infection. The subjects received the vector vaccine (ChAdOx1 nCoV-19, AstraZeneca, Cambridge, UK), MM vaccine (1st vaccination: ChAdOx1 nCoV-19; 2nd vaccination mRNA vaccine), or mRNA vaccine (BNT162b2, Pfizer-BioNTech, Pfizer Inc., New York, NY, USA; mRNA-1273, Moderna, Inc., Cambridge, MA, USA). Subjects were excluded if they: (1) were less than 18 years old, (2) had a history of SARS-CoV-2 infection, or (3) were on medication with immunosuppressants. All subjects were provided written informed consent before enrollment and a questionnaire at every blood collection. The questionnaire inquired about current COVID-19 symptoms, a history of prior infection, contact with confirmed patients, adverse responses following vaccination, and underlying disease. Enrolled subjects were tested for N Ab, RBD Ab, or nAb to monitor unknown natural infection. Despite having no history of natural infection, subjects with clinically suspect symptoms underwent reverse transcription-polymerase chain reaction to confirm SARS-CoV-2 infection. Subjects exhibiting positive results were subsequently excluded. The serum samples for each vaccinated subject were obtained three times: T0 for pre-vaccination, T1 for time points of 3–4 weeks (mRNA vaccine) or 7–8 weeks (vector vaccine) after their 1st vaccination, and T2 for the time point of 3–4 weeks after their 2nd vaccination. This study included 1851 serum samples obtained from 733 vaccinated subjects (vector, 484 subjects; MM, 145 subjects; mRNA 104 subjects). This study was approved by the Institutional Review Board of KUAH (K2021-0511-014). Informed consent was obtained from all vaccinated subjects before sample collection. 

### 2.2. SARS-CoV-2 IgG Antibody Binding Assays

The vaccinated group was evaluated using Alinity SARS-CoV-2 IgG II Quant (Abbott Laboratories, Sligo, Ireland) (RBD Ab_A), Access SARS-CoV-2 IgG (BeckmanCoulter Inc., CA, USA) (RBD Ab_B), and Elecsys anti-SARS-CoV-2 S (Roche Diagnostics GmbH, Mannheim, Germany) (RBD Ab_R) for RBD Ab assays according to the manufacturer’s instructions. RBD Ab_A, RBD Ab_B, and RBD Ab_R measured IgG Abs against the RBD of the spike protein in SARS-CoV-2. RBD Ab_A, RBD Ab_B, and RBD Ab_R were designed to provide qualitative and quantitative determination of SARS-CoV-2 IgG Abs. RBD Ab_A is an automated two-step sandwich immunoassay using indirect chemiluminescent microparticle technology. Antibody concentration is presented using arbitrary units (AU/mL). Arbitrary units were converted to binding antibody units per milliliter (BAU/mL), which are WHO international standards for anti-SARS-CoV-2 immunoglobulin, using the following equation: BAU/mL = 0.142 × AU/mL. The cutoff for the determination of a negative or positive value was 7.1 BAU/mL. The analytical measurement interval (AMR) is 3.0–5680 BAU/mL, and the extended measuring interval (EMI) by dilution was 5680–11,360 BAU/mL. RBD Ab_B is an automated two-step enzyme immunoassay using a magnetic particle-chemiluminescence enzyme technology. The antibody concentration was determined as a non-reactive or reactive response based on a cutoff of 30 BAU/mL. The AMR is 8.0–1800 BAU/mL and the EMI is 1500–36,000 BAU/mL. RBD Ab_R is an automated two-step sandwich immunoassay using electrochemiluminescence technology. The antibody concentration is presented using units of U/mL and was converted to BAU/mL using the following equation: BAU/mL = 0.972 × U/mL. The cutoff for antibody concentration was 0.8 BAU/mL. The AMR is 0.4–243.0 BAU/mL, and results over 243 BAU/mL were diluted and re-tested to obtain accurate results. 

### 2.3. SARS-CoV-2 Neutralizing Antibody Assay

The vaccinated group was evaluated using the GenScript for the nAb assay according to the manufacturer’s instructions. The nAb values were measured by competitive ELISA. The ACE2 receptor precoated on the microplate was incubated with horseradish peroxidase(HRP)-labeled RBD, producing a strong signal. If neutralizing antibodies were present in the sample, they would attach to HRP-labeled RBD and protect it from binding to the microplate’s ACE2 receptor. A low signal is produced by serum samples containing more nAb. The signal (optical density, OD) was measured at 450 nm on a spectrophotometer and was calculated using the following formula: signal inhibition (%) = (1 − OD value of sample/OD value of negative control) × 100. The signal inhibition results were converted to IU/mL according to the formula suggested by the manufacturer. The nAb values were determined to be a negative or positive response based on a cutoff of ≥28 IU/mL. 

### 2.4. Interferon-Gamma Release Assay for SARS-CoV-2

The vaccinated group was evaluated using the QuantiFERON for the IGR assay (Qiagen, Venlo, The Netherlands, research use only) according to the manufacturer’s instructions. Each 1 mL blood sample was directly placed into four QuantiFERON SARS-CoV-2 blood collection tubes (Qiagen): a nil tube, an Ag1 tube containing CD4^+^ epitopes of the S1 subunits (class II), an Ag2 tube containing CD4^+^ T cell and CD8^+^ T cell-specific epitopes (class I/II) of the S1 and S2 subunits, and the mitogen tube. The nil tube was used to calculate the background signal, and the mitogen tube was used as a positive control. IGR values were determined by subtracting the values for the nil tube from those for the class II or class I/II tube. The manufacturer did not suggest a definite cutoff. We determined the cutoff by a receiver operating characteristic curve where the cutoff of class II was >0.037 IU/mL and that of class I/II was >0.052 IU/mL. 

### 2.5. Statistical Analysis

All data were evaluated for a normal distribution and homogenous variation using the Kolmogorov–Smirnov test. Continuous variables with a non-parametric distribution are presented as the median (interquartile range, IQR), and categorical variables with a non-parametric distribution are presented as numbers (percentages). The values of the RBD Ab_A, RBD Ab_B, RBD Ab_R, nAb, and IGR assays were evaluated and compared depending on the vaccine type including vector, MM, and mRNA vaccines. The qualitative values among the RBD Ab_A, RBD Ab_B, RBD Ab_R, nAb, and IGR assays were evaluated using concordance rates including total agreement (TA), positive agreement (PA), negative agreement (NA), and the Cohen’s kappa coefficient with a 95% confidence interval (CI) with the following suggestion: 0.00–0.20, none; 0.21–0.39, minimal; 0.40–0.59, weak; 0.60–0.79, moderate; 0.80–0.90, strong, >0.90, almost perfect agreement [23]. The qualitative values among the RBD Ab_A, RBD Ab_B, RBD Ab_R, nAb, and IGR assays were analyzed using Spearman’s correlation coefficient (ρ) and Passing–Bablok regression. The two-sided 95% CI of the intercept and slope were calculated. The correlation coefficient was interpreted by the following suggestion: <0.20, negligible; 0.20–0.29, weak; 0.30–0.39, moderate; 0.40–0.69, strong, 0.70–0.99, very strong, and 1.00, perfect [24]. All statistical analyses were conducted using MedCalc version 20.014 (MedCalc Software Bvba, Ostend, Belgium).

## 3. Results

### 3.1. Comparison of Qualitative and Quantitative Values in the Vaccinated Group 

The median values of the RBD Ab, nAb, and IGR assays were sequentially increased after T1 and T2 (Table 1). The TA values were over 79.4% (range 79.4–98.4%) (Table 2). The agreement between RBD Ab_A and RBD Ab_R was almost perfect (kappa 0.96, 95% CI 0.95–0.98). Among three RBD Ab assays, RBD Ab_B showed the highest agreement with the nAb assay (kappa 0.83, 95% CI 0.80–0.85) and the lowest agreement with the IGR assay (kappa 0.55, 95% CI 0.51–0.60). Agreement between the nAb assay and the IGR assay was weak (kappa 0.58, 95% CI 0.54–0.63). The quantitative values among RBD Ab assays and nAb assays showed very strong associations (rho range 0.86–0.95) and those between the IGR assay and other assays showed strong to very strong associations (rho range 0.57–0.96) (Figure 1).

### 3.2. Comparison of Qualitative and Quantitative Values according to the Vaccine Types

The median values of the RBD Ab, nAb, and IGR assays were higher in the MM vaccine group or mRNA vaccine group than in the vector vaccine group (Table 1). The ranges of the TA values were 75.4%–98.3% in the vector vaccine group, 80.8%–98.2% in the MM vaccine group, and 89.4%–99.6% in the mRNA vaccine group (Table 3). The agreements were the highest in the mRNA vaccine group (kappa range 0.76–0.99) (Table 3). The agreements were relatively low between the IGR assay and other assays (kappa range 0.48–0.79). Regarding the quantitative values, the ranges of the rho correlation coefficient were 0.39–0.96 in the vector vaccine group, 0.62–0.98 in the MM vaccine group, and 0.78–0.96 in the mRNA vaccine group (Figure 2). The rho correlation coefficients between the RBD Ab assays and nAb assay were higher in the MM vaccine group and those between IGR and other assays were higher in the mRNA vaccine group (Table 4).

## 4. Discussion

There are currently numerous anti-SARS-CoV-2 antibody immunoassays on the market. This can be overwhelming for laboratory directors who need to choose which test to use in routine practice. At the start of the pandemic, these immunoassays were primarily used to confirm SARS-CoV-2 infection and there were debates about which tests showed better performance based on N or S (RBD) proteins [25,26]. The rollout of vaccines in late 2020 resulted in a preference for RBD Ab assays in assessing the humoral immune response to the COVID-19 vaccine and its role in estimating protective immunity became important [27]. Regarding protective immunity, nAb and antigen-specific memory T-cells are critical parameters. The nAb is a key player in the humoral immune responses that defends blocking viral entry into host cells and neutralizing their biological effect in collaboration with immune cells [8]. Apart from the humoral immune responses, effective defense against SARS-CoV-2 can be achieved through cellular immune responses, including T cells and their related memory subsets [28]. Thus, considering the successful prevention of viral infection and spreading, the type of vaccine, role of RBD Ab, and associations between humoral immune responses and cellular immune responses should be demonstrated. 

In this study, we evaluated the qualitative and quantitative values among the RBD assay, nAb assay, and IGR assay in the vaccinated group. The values were compared based on the vaccine type, vector vaccine, MM vaccine, and mRNA vaccine, in a large population. In the qualitative and quantitative values of each assay, RBD Ab assays showed better performance in predicting the presence of nAb activity, but not the IGR level. The quantitative and qualitative values showed the lowest agreement and correlation in the vector vaccine group. The correlations between the RBD Ab assays and nAb assay were very strong in the vector vaccine, MM vaccine, and mRNA vaccine groups, and those between the RBD assays and IGR assay were very strong in only the mRNA vaccine group. 

The virus neutralization test is considered the gold-standard method to quantify nAb values against SARS-CoV-2 [10]. However, this method is time-consuming and labor-intensive, and requires biosafety laboratory level 3 facilities to handle risk group 3 pathogens such as SARS-CoV-2 [10]. sVNT is a commercially available assay and showed a good correlation with the gold-standard method and the nAb titer [9,29]. However, the TA values and correlation results were very different depending on the RBD assay platform, the number of vaccinations, and vaccine type [4,5]. In this study, the RBD Ab assays and nAb assay showed moderate to strong agreement (kappa 0.76–0.83), and correlation coefficients were strong to very strong (ρ = 0.88–0.94). Similar to this study, a previous comparison study between the RBD Ab assays and nAb assays showed moderate to strong agreement (kappa 0.80–0.84) and a very strong correlation (correlation coefficient 0.89–0.90) [21]. However, for international standardization to allow accurate calibration of assays and to use common units for data, the number of binding antibody units (BAU) per mL is used to compare assays detecting the same class of immunoglobulins with the same specificity [30]. Since the conversion formula for each assay used a constant value suggested by the manufacturers, systemic bias might occur and affect the results. 

We confirmed that the IGR values modestly increased after vaccination, but the agreement and correlation were relatively low between the IGR assay and other assays. In previous reports, the correlation between the RBD Ab and IGR values showed discrepancies [21,22]. First, the cellular immune responses after vaccination greatly varied from individual to individual due to pre-existing cross-reactive memory T-cells against SARS-CoV-2 [13]. Some studies have demonstrated that reactive T cells for SARS-CoV-2 are present in as many as 60% of individuals who have not been exposed to SARS-CoV-2. This implies that there might be a cross-reactive T-cell recognition between the coronavirus that causes the common cold and SARS-CoV-2 [14,31].

Regarding vaccine type, this study showed consistent findings with previous studies. The values of RBD Ab, nAb, and IGR in the mRNA vaccine group were higher than in vector and MM vaccines [4,32,33,34]. The qualitative and quantitative values were higher in mRNA with high Ab values than in vector and MM vaccines with relatively low Ab values. In particular, the agreement and correlation between the RBD Ab assays and IGR assay and between the RBD Ab assays and nAb assay were higher in the mRNA vaccine than in the vector and MM vaccine groups. With the superb correlation, the higher the RBD Ab value, the better agreement and correlation with the nAb values [5,19,35]. Thus, this study suggests the high potential of RBD Ab assays for qualitatively and quantitatively predicting IGR in an individual with the mRNA vaccine showing high values of RBD Ab. 

This study has some limitations. First, our assay utilized a recombinant spike RBD based on the original SARS-CoV-2 strain. Thus, antibody binding avidity to recently emerged variants such as delta or omicron could be lower than the values we measured in this study. Second, this is a cross-sectional study. We do not consider the influence of Ab values decay dynamics according to time intervals. Despite these limitations, this study demonstrated that the RBD Ab assays showed good performance in predicting the presence of nAb activity. IGR level can be predicted by RBD assays only in the mRNA vaccine group. 

In conclusion, different RBD Ab assays showed comparable performance and good correlation with the nAb assay across vaccine types. The RBD assays indicated superb agreement and correlation with IGR assay in the mRNA vaccine group with high Ab values. This study provides a better understanding of serological assays and the IGR assay for SARS-CoV-2 in the vaccinated group with a broad range of Ab values. 

## Figures and Tables

**Figure 1 diagnostics-13-03688-f001:**
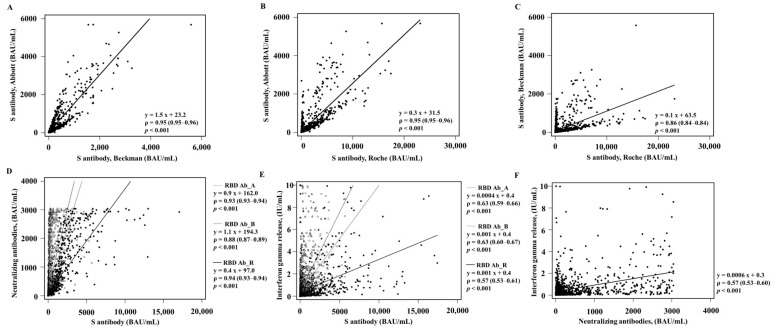
Quantitative correlation among the RBD Ab assays and nAb assay. (**A**) RBD Ab_A vs. RBD Ab_B, (**B**) RBD Ab_A vs. RBD Ab_R, and (**C**) RBD Ab_B vs. RBD Ab_R. The black line indicates the regression line of the vaccinated group. (**D**) RBD Ab_A, (grey triangle and dotted grey line) (**E**) RBD Ab_B (empty circle and solid grey line), and (**F**) RBD Ab_R (filled circle and solid black line) were compared with the nAb assay. Abbreviations: RBD Ab, receptor-binding domain antibodies; BAU, binding antibody units.

**Figure 2 diagnostics-13-03688-f002:**
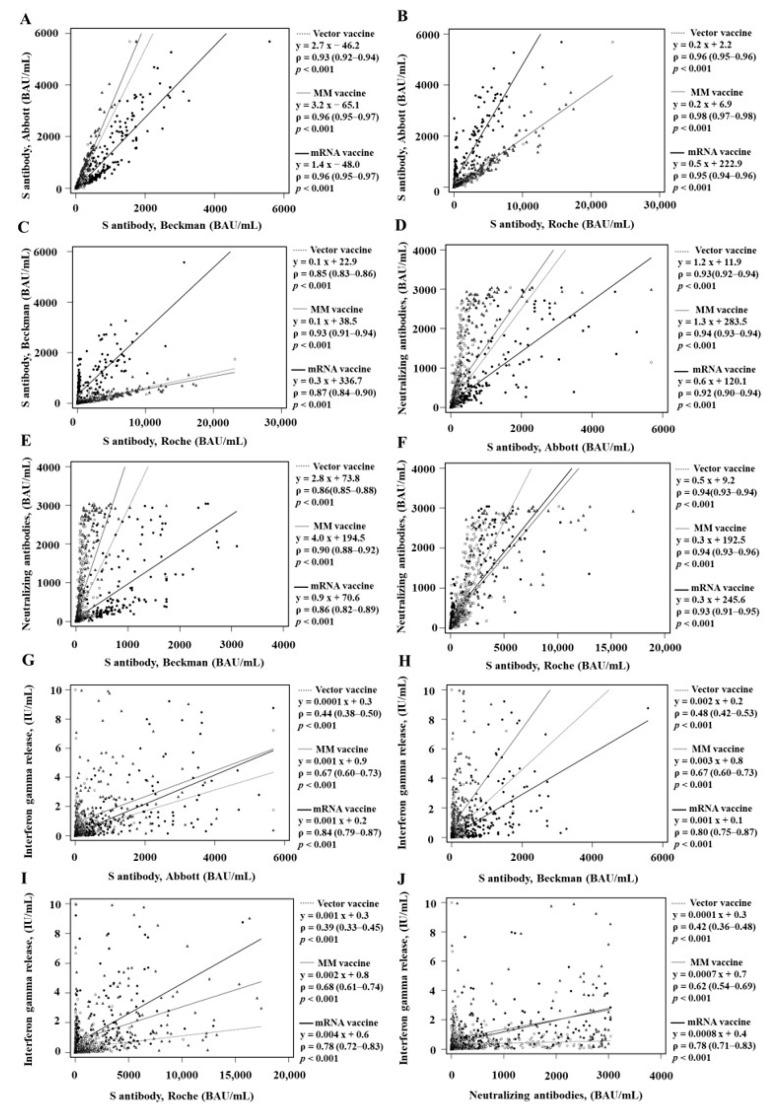
Quantitative correlation according to vaccine type among the RBD Ab assays and nAb assay. (**A**) RBD Ab_A vs. RBD Ab_B, (**B**) RBD Ab_A vs. RBD Ab_R, and (**C**) RBD Ab_B vs. RBD Ab_R were compared according to vaccine among BRD Ab assays. nAb assay was compared with (**D**) RBD Ab_A, (**E**) RBD Ab_B, and (**F**) RBD Ab_R. IGR assay was compared with (**G**) RBD Ab_A, (**H**) RBD Ab_B, (**I**) RBD Ab_R, and (**J**) the nAb assay. The regression lines were presented by the grey dotted line for vector vaccine, the grey solid for MM vaccines, and the black solid line for mRNA vaccine groups.

**Table 1 diagnostics-13-03688-t001:** Antibody and IGR levels for the RBD Ab assay, nAb assay, and IGR assay according to sample collection time and vaccine type.

	Total (*n* = 1851)	Vector Vaccine (*n* = 1264)	MM Vaccine (*n* = 338)	mRNA Vaccine (*n* = 249)	*p*
RBD Ab_A (BAU/mL)					
Total	25.0 (0.0–117.9)	22.4 (0.0–83.1)	24.9 (0.0–286.5)	133.6 (0.0–864.8)	<0.001
T_0_	0.0 (0.0–0.0)	0.0 (0.0–0.0)	0.0 (0.0–0.0)	0.0 (0.0–0.0)	0.915
T_1_	36.2 (9.4–83.2)	33.8 (11.2–61.5)	25.5 (0.0–54.8)	394.1 (163.2–629.4)	<0.001
T_2_	149.5 (63.5–468.4)	101.1 (48.1–197.4)	654.7 (273.6–1164)	1506 (74.8–2796)	<0.001
RBD Ab_B (BAU/mL)					
Total	36.8 (1.8–93.6)	29.8 (1.8–65.4)	60.0 (2.3–198.0)	472.8 (1.5–1056)	<0.001
T_0_	1.0 (0.6–1.7)	1.0 (0.6–1.7)	0.9 (0.6–1.6)	0.9 (0.6–1.6)	0.432
T_1–_	56.1 (29.4–110.6)	43.2 (25.5–74.5)	56.3 (29.6–93.9)	618.9 (462.5–860.9)	<0.001
T_2_	75.4 (42.7–198.0)	56.0 (32.8–90.4)	214.6 (95.5–394.9)	1318 (654.7–1738)	<0.001
RBD Ab_R (BAU/mL)					
Total	31.1 (0.0–411.9)	33.5 (0.0–318.1)	20.5 (0.0–1766)	64.7 (0.0–460.5)	0.044
T_0_	0.0 (0.0–0.0)	0.0 (0.0–0.0)	0.0 (0.0–0.0)	0.0 (0.0–0.0)	0.132
T_1_	44.3 (10.0–93.2)	43.6 (14.3–81.3)	21.1 (0.0–51.1)	152.1 (60.8–219.2)	<0.001
T_2_	982.7 (391.2–2516)	678.5 (237.9–1307)	3765 (1766–6099)	2869 (174–4420)	<0.001
nAb (IU/mL)					
Total	39.5 (5.0–238.	33.6 (4.6–134.8)	47.6 (9.0–1096)	208.3 (0.9–619.5)	<0.001
T_0_	1.2 (0.0–6.1)	1.4 (0.0–5.8)	5.3 (0.0–9.6)	0.0 (0.0–2.0)	<0.001
T_1_	47.4 (22.5–108.9)	39.5 (19.5–81.6)	37.5 (19.5–81.6)	276.4 (188.5–481.5)	<0.001
T_2_	416.1 (126.0–1473)	235.8 (90.0–653.1)	1861 (706.5–2713)	1791 (1091–2506)	<0.001
IGRA, class I (IU/mL)					
Total	0.09 (0.00–0.37)	0.07 (0.01–0.23)	0.20 (−0.00–0.91)	0.10 (0.00–0.89)	
T_0_	−0.00 (−0.02–0.01)	0.00 (−0.02–0.01)	−0.01 (−0.05–0.01)	0.00 (−0.01–0.01)	0.020
T_1_	0.14 (0.04–0.42)	0.11 (0.04–0.30)	0.18 (0.07–0.68)	0.22 (0.07–0.72)	<0.001
T_2_	0.21 (0.07–0.65)	0.14 (0.05–0.30)	0.67 (0.24–1.90)	1.14 (0.44–2.88)	<0.001
IGRA, class I/II (IU/mL)					
Total	0.16 (0.01–0.67)	0.13 (0.02–0.41)	0.30 (0.01–0.58)	0.20 (0.01–1.48)	
T_0_	0.00 (−0.01–0.02)	0.00 (−0.01–0.02)	0.00 (−0.03–0.02)	0.00 (−0.00–0.01)	0.446
T_1_	0.27 (0.10–0.85)	0.19 (0.07–0.56)	0.50 (0.16–1.65)	0.41 (0.15–1.04)	0.001
T_2_	0.37 (0.11–1.15)	0.24 (0.09–0.55)	1.25 (0.40–3.32)	2.08 (0.79–4.00)	<0.001

All data are represented as the median (interquartile range). Abbreviations: IGR, interferon-gamma release; RBD Ab, receptor-binding domain antibodies; nAb, neutralizing antibody; MM, mix-and-match; *n*, number; BAU, binding antibody units; RBD Ab_A, receptor-binding domain antibodies using Abbott kit; RBD Ab_B, receptor-binding domain antibodies using Beckman kit; RBD Ab_R, receptor-binding domain antibodies using Roche kit; nAb, neutralizing antibody; IGRA, interferon-gamma release assay.

**Table 2 diagnostics-13-03688-t002:** Concordance rates and agreements among the RBD Ab assays, nAb assay, and IGR assay.

	TA Case *n*/Total *n* % (95% CI)	PA Case *n*/Total *n* % (95% CI)	NA Case *n*/Total *n* % (95% CI)	Kappa (95% CI)
RBD Ab_A vs. RBD Ab_B	87.3 (83.1–91.7)	81.3 (76.3–86.5)	99.4 (91.7–100)	0.74 (0.71–0.77)
RBD Ab_A vs. RBD Ab_R	98.4 (93.9–100)	99.5 (94.0–100)	96.1 (88.5–100)	0.96 (0.95–0.98)
RBD Ab_B vs. RBD Ab_R	86.2 (82.1–90.6)	99.5 (93.4–100)	70.4 (64.9–76.3)	0.72 (0.68–0.75)
RBD Ab_A vs. nAb	89.4 (85.2–93.8)	99.7 (93.8–100)	76.1 (70.2–82.4)	0.78 (0.75–0.81)
RBD Ab_B vs. nAb	91.4 (87.1–95.9)	90.6 (84.9–96.6)	92.5 (85.9–99.3)	0.83 (0.80–0.85)
RBD Ab_R vs. nAb	88.4 (84.2–92.8)	99.7 (93.8–100)	73.9 (68.1–80.1)	0.76 (0.73–0.79)
RBD Ab_A vs. IGR	91.5 (90.0–92.8)	91.7 (90.2–93.0)	72.4 (70.1–74.6)	0.66 (0.62–0.70)
RBD Ab_B vs. IGR	79.4 (77.3–81.4)	78.5 (76.3–80.5)	81.6 (79.5–83.5)	0.55 (0.51–0.60)
RBD Ab_R vs. IGR	86.2 (84.4–87.9)	92.6 (91.2–93.8)	71.7 (69.4–73.9)	0.66 (0.62–0.71)
nAb vs. IGR	87.6 (85.8–89.2)	90.0 (88.4–91.4)	66.3 (63.9–68.7)	0.58 (0.54–0.63)

Abbreviations: TA, total agreement; PA, positive agreement; NA, negative agreement; *n*, number; CI, confidence interval; RBD Ab_A, receptor-binding domain antibodies using Abbott kit; RBD Ab_B, receptor-binding domain antibodies using Beckman kit; RBD Ab_R, receptor-binding domain antibodies using Roche kit; neutralizing antibody; IGRA, interferon-gamma release assay.

**Table 3 diagnostics-13-03688-t003:** Concordance rates and agreement among RBD assays according to the vaccine types.

Assay	Vector Vaccine	MM Vaccine	Mrna Vaccine
RBD Ab_A vs. RBD Ab_B			
TA% (95% CI)	83.9 (78.9–89.1)	91.1 (81.2–100)	99.6 (87.6–100)
PA% (95% CI)	75.7 (69.9–81.8)	88.2 (76.6–100)	99.4 (84.8–100)
NA% (95% CI)	99.5 (90.4–100)	98.0 (79.7–100)	100 (79.7–100)
Kappa (95% CI)	0.68 (0.64–0.72)	0.80 (0.74–0.87)	0.99 (0.97–1.00)
RBD Ab_A vs. RBD Ab_R			
TA% (95% CI)	98.3 (93.0–100)	98.2 (87.9–100)	99.2 (87.2–100)
PA% (95% CI)	99.8 (93.1–100)	99.2 (86.9–100)	98.8 (84.3–100)
NA% (95% CI)	95.4 (86.4–100)	96.0 (77.9–100)	100 (79.7–100)
Kappa (95% CI)	0.96 (0.95–0.98)	0.96 (0.92–0.99)	0.98 (0.96–1.00)
RBD Ab_B vs. RBD Ab_R			
TA% (95% CI)	82.6 (77.7–87.8)	90.5 (80.7–100)	98.8 (86.8–100)
PA% (95% CI)	99.8 (92.2–100)	99.1 (86.1–100)	98.2 (83.7–100)
NA% (95% CI)	95.6 (86.6–1.05)	76.4 (61.9–93.2)	98.8 (78.7–100)
Kappa (95% CI)	0.65 (0.61–0.69)	0.79 (0.72–0.86)	0.97 (0.94–1.00)
RBD Ab_A vs. nAb			
TA% (95% CI)	86.9 (84.9–88.7)	91.4 (87.9–93.9)	99.2 (97.1–99.8)
PA% (95% CI)	99.7 (99.2–99.9)	100 (98.9–100)	98.8 (94.5–99.6)
NA% (95% CI)	72.6 (70.1–75.0)	77.7 (73.0–81.8)	99.4 (97.4–99.9)
Kappa (95% CI)	0.73 (0.70–0.77)	0.81 (0.75–0.88)	0.98 (0.96–1.00)
RBD Ab_B vs. nAb			
TA% (95% CI)	89.3 (87.5–90.9)	93.8 (90.7–95.9)	98.8 (96.5–99.6)
PA% (95% CI)	87.0 (85.0–88.7)	95.7 (93.0–97.4)	98.8 (96.5–99.6)
NA% (95% CI)	91.9 (90.3–93.3)	90.8 (87.2–93.4)	98.8 (96.5–99.6)
Kappa (95% CI)	0.79 (0.75–0.82)	0.87 (0.81–0.92)	0.97 (0.94–1.00)
RBD Ab_R vs. nAb			
TA% (95% CI)	85.8 (83.8–87.6)	90.2 (86.6–92.9)	99.2 (97.1–99.8)
PA% (95% CI)	100 (99.2–100)	99.5 (98.0–99.9)	98.8 (96.5–99.6)
NA% (95% CI)	69.9 (67.3–72.4)	75.4 (70.5–79.7)	100 (98.5–100)
Kappa (95% CI)	0.71 (0.67–0.75)	0.78 (0.72–0.85)	0.98 (0.96–1.00)
RBD Ab_A vs. IGR			
TA% (95% CI)	83.9 (81.4–86.2)	87.7 (83.7–90.8)	89.8 (85.4–93.0)
PA% (95% CI)	92.1 (90.2–93.7)	90.4 (868–93.1)	92.1 (88.1–94.9)
NA% (95% CI)	65.7 (62.5–68.7)	80.9 (763–84.8)	85.4 (80.4–89.3)
Kappa (95% CI)	0.61 (0.55–0.66)	0.70 (0.62–0.79)	0.77 (0.68–0.86)
RBD Ab_B vs. IGR			
TA% (95% CI)	75.4 (72.5–78.1)	82.9 (78.5–86.6)	89.4 (84.9–92.7)
PA% (95% CI)	73.8 (70.8–76.6)	81.7 (77.2–85.5)	91.5 (87.4–94.4)
NA% (95% CI)	78.9 (76.1–81.4)	86.2 (82.1–89.5)	85.4 (80.4–89.3)
Kappa (95% CI)	0.48 (0.42–0.54)	0.62 (0.53–0.70)	0.76 (0.68–0.85)
RBD Ab_R vs. IGR			
TA% (95% CI)	84.2 (81.7–86.4)	88.3 (84.4–91.3)	90.7 (86.4–93.7)
PA% (95% CI)	93.3 (94.5–94.8)	91.3 (87.8–93.9)	92.1 (88.1–94.9)
NA% (95% CI)	63.9 (60.7–67.0)	80.9 (76.3–84.8)	87.8 (83.1–91.3)
Kappa (95% CI)	0.61 (0.55–0.66)	0.71 (0.63–0.80)	0.79 (0.71–0.87)
nAb vs. IGR			
TA% (95% CI)	79.1 (76.3–81.6)	80.8 (76.2–84.7)	89.8 (85.4–93.0)
PA% (95% CI)	76.1 (73.2–78.8)	79.6 (75.0–83.6)	92.1 (88.1–94.9)
NA% (95% CI)	80.4 (77.7–82.9)	84.0 (79.7–87.5)	85.4 (80.4–89.3)
Kappa (95% CI)	0.54 (0.48–0.59)	0.57 (0.48–0.66)	0.77 (0.69–0.86)

Abbreviations: See Table 1 and Table 2.

**Table 4 diagnostics-13-03688-t004:** Mean difference of the values of antibody or interferon gamma release among RBD Ab_A, RBD Ab_B, RBD Ab_R, nAb, and IGR.

Assay	Mean Differences			
	Total	Vector Vaccine	MM Vaccine	mRNA Vaccine
RBD Ab_A vs. RBD Ab_B	99.5 (83.6–115.4)	34.9 (25.0–44.7)	253.6 (198.1–309.2)	218.3 (148.4–288.2)
RBD Ab_A vs. RBD Ab_R	−481.3 (−540.2–−422.4)	−279.0 (−318.3–−239.8)	−1244 (−1479–−1010)	−362.7 (−508.2–−217.2)
RBD Ab_B vs. RBD Ab_R	−666.6 (−748.2–−584.9)	−352.2 (−403.9–−300.5)	−1841 (−2180–−1502)	−671.3 (−903.9–438.7)
RBD Ab_A vs. nAb	−136.9 (−160.7–−113.2)	−126.7 (−148.4–−105.0)	−369.7 (−447.9–−291.4)	127.9 (45.5–210.3)
RBD Ab_B vs. nAb	−208.9 (−236.1–−181.8)	−156.6 (−180.2–−133.1)	−559.6 (−660.1–−459.0)	−11.1 (−89.0–−66.8)
RBD Ab_R vs. nAb	326.2 (272.5–379.9)	179.6 (138.3–220.8)	837.9 (625.2–1051)	424.0 (240.4–607.6)
RBD Ab_A vs. IGR	302.6 (266.8–338.4)	104.3 (845.0–123.5)	400.1 (323.1–477.1)	898.1 (746.5–1050)
RBD Ab_B vs. IGR	177.7 (156.6–198.8)	54.4 (48.2–60.6)	144.0 (121.3–166.7)	676.3 (576.0–776.5)
RBD Ab_R vs. IGR	1005 (895.4–1115)	543.8 (469.2–618.3)	1997 (1638–2357)	1349 (1048–1651)
nAb vs. IGR	410.1 (371.9–448.4)	272.6 (238.4–306.7)	685.3 (570.0–800.6)	571.9 (462.8–681.0)

Abbreviations: See Table 1 and Table 2.

## Data Availability

A dataset of 2197 samples was deposited at https://dataverse.harvard.edu/ (https://doi.org/10.7910/DVN/4FUNZA) (accessed on 13 December 2023).

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
