# Peer review of "Performance Evaluation of Three Antibody Binding Assays, a Neutralizing Antibody Assay, and an Interferon-Gamma Release Assay for SARS-CoV-2 According to Vaccine Type in Vaccinated Group"

_diagnostics, 2023, doi:10.3390/diagnostics13243688_

Round 1

Reviewer 1 Report

Comments and Suggestions for Authors

lines 66, 67 and 79, described a "S Ab", but miss the meaning and were not used elsewhere in the article.

in material and methods it is not clear whether the tests were done in accordance  to the kit protocol.

Tables should be in the supplementary materials

Author Response

  1. Line 66, 67, and 79, described as “S Ab”, but miss the meaning and were not used elsewhere in the article.

Thank you for your comment. We modified “S Ab” to RBD antibody.  

Many studies have evaluated the performance of RBD antibody (Ab) assays and the nAb assay among vaccinated groups. Depending on the RBD Ab assay platform, RBD Ab assays had a sensitivity of 90.0% to 97.4% and a specificity of 97.9% to 100%, [15]. Moreover, the RBD Ab assay and nAb assay were associated to varying degrees [16-19]. Unlike commercial assays measuring humoral immune responses, the conventional methods of detecting cellular immune responses are complex and have not yet been standardized, requiring highly specialized facilities [20-22]. Moreover, few studies comprehensively compared the performance of RBD Ab assay, nAb assay, and IGR assay for SARS-CoV-2 in vaccinated group. (page 2, line 72)

  1. In material and methods, it is not clear whether the tests were done in accordance to the kit protocol..

Thank you for your valuable comment. We added the following sentences in the material and method section.

The vaccinated group was evaluated using Alinity SARS-CoV-2 IgG II Quant (Abbott Laboratories, Sligo, Ireland) (RBD Ab_A), Access SARS-CoV-2 IgG (BeckmanCoulter Inc., CA, USA) (RBD Ab_B), and Elecsys anti-SARS-CoV-2 S (Roche Diagnostics GmbH, Mannheim, Germany) (RBD Ab_R) for RBD Ab assays according to the manufacturer’s in-structions. (page 3, line 100)

The vaccinated group was evaluated using the GenScript for the nAb assay according to the manufacturer’s instructions. (page 3, line 125)

The vaccinated group was evaluated using the QuantiFERON for the IGR assay (Qiagen, Venlo, The Netherlands, research use only) according to the manufacturer’s instructions. (page 3, line 138)

  1. Tables should be in the supplementary materials.

Thank you for your valuable comment. We carefully considered your suggestion regarding the placement of tables in the supplementary material. We appreciate your concern for the manuscript's conciseness and clarity. However, after thorough consideration, we believe that the inclusion of the four tables in the main article is crucial for providing comprehensive support to our conclusions. These tables play a pivotal role in presenting essential data that directly contributes to the robustness of our findings. Placing them in the supplementary material may risk diminishing the overall accessibility of our results and may hinder the reader's ability to correlate specific details with the main narrative. Furthermore, the information contained in these tables is integral to the coherence and completeness of our argumentation. We appreciate your valuable guidance.

Reviewer 2 Report

Comments and Suggestions for Authors

In the manuscript entitled ''Performance Evaluation of Three Antibody Binding Assays, a Neutralizing Antibody Assay, and an Interferon-Gamma Release Assay for SARS-CoV-2 according to Vaccine Type in Vaccinated Group'' Nam et al. tried to compare different commercial assays and extrapolated their performance on samples from individuals immunized against SARS-CoV-2.

Major concern:

Authors have not considered that subjects can develop natural infection during immunization scheme. None of the subjects was examined for presence of COVID-19 related signs and symptoms.

Also, excluding and including criteria for subjects was not mentioned.

With manuscript in current state, we cant have any objective conslusions.

Minor comments are in the file attached.

Author Response

  1. line 41. Be more specific.

Thank you for your comment. According to your comment, we modified the following sentence.

Immunization emerges as the preeminent and secure strategy for mitigating the impact of the ongoing pandemic, demonstrating unparalleled safety and efficacy in curtailing the spread of the infectious agent and averting severe clinical outcomes. (page 1, line 41)

  1. line 43, line 47. And not through natural infection?

Thank you for your comment. According to your comment, we modified the following sentence.

The population acquires humoral or cellular immune responses against SARS-CoV-2 through vaccination and natural infection. (page 2, line 45)

The Ab assay measures the humoral immune responses after vaccination and natural infection and has been developed for different target antigens and assay platforms. (page 2, line 49)

  1. line 50. Be more specific.

Thank you for your comment. According to your comment, we modified the following sentence.

Many Ab assays with high sensitivity and specificity can be harmonized and validated using World Health Organization (WHO) International Standard (IS) [7]. (page 2, line 53)

We added the relevant reference.

  1. Kemp TJ, Hempel HA, Pan Y, RoyD, Cherry J, Lowy DR, Pinto LA (2023) Assay harmonization study to measure immune response to SARS-CoV-2 infection and vaccines: a serology methods study. Microbiol Spectr. 11:e05353-22. doi: 10.1128/spectrum.05353-22 (page 11, line 335)

  1. line 53. Not true. Neutralization assay can’t predict is someone is resistant to SARS-CoV-2 infection, since it is completely excluding cellular immunity.

Thank you for your comment. According to your comment, we modified the following sentence.

The nAb assay also measures the humoral immune responses that defend against blocking viral entry into host cells and neutralizing their biological effect in collaboration with immune cells [8]. (page 2, line 55)

  1. line 58. What about other interactions?

Thank you for your comment. According to your comment, we modified the following sentence.

A commercialized nAb assay assesses potential values as a surrogate for nAb that blocks the interactions between RBD and the angiotensin-converting enzyme 2 (ACE2) receptor [9]. (page 2, line 57)

  1. line 59. State why they are urgently needed.

Thank you for your comment. According to your comment, we modified the following sentence.

However, assessing the cellular immune responses is essential for understanding long-term immunity, vaccine effectiveness, and vaccine durability as memory T cells play a key role in providing sustained protection beyond the humoral immune responses [11]. (page 2, line 63)

  1. line 68. Be more precise, what is excellent?

Thank you for your comment. According to your comment, we modified the following sentence.

Depending on the RBD Ab assay platform, RBD Ab assays had a sensitivity of 90.0% to 97.4% and a specificity of 97.9% to 100%, [15]. Moreover, the RBD Ab assay and nAb assay were associated to varying degrees [16-19]. (page 2, line 73)

We added the relevant reference.

  1. Chen SY, Lee YL, Lin YC, Lee NY, Liao CH, Hung YP, Lu MC, Wu JL, Tseng WP, Lin CH, Chung MY, Kang CM, Lee YF, Lee TF, Cheng CY, Chen CP, Huang CH, Liu CE, Cheng SH, Ko WC, Hsueh PR, Chen SC (2020) Multicenter evaluation of two chemiluminescence and three lateral flow immunoassays for the diagnosis of COVID-19 and assessment of antibody dynamic responses to SARS-CoV-2 in Taiwan. Emerg Microbes Infect 9:2157–2168. doi: 10.1080/22221751.2020.1825016. (page 11, line 360)

  1. line 70. Explain more precise. Which performance you are referring to?

Thank you for your comment. According to your comment, we modified the following sentence.

Unlike commercial assays measuring humoral immune responses, the conventional methods of detecting cellular immune responses are complex and have not yet been standardized, requiring highly specialized facilities [20-22]. (page 2, line 75)

  1. line 85. Be more precise, there are two Koreas.

Thank you for your comment. According to your comment, we modified the following sentence.

This study was conducted from March 2021 to May 2022 at the Korea University Anam Hospital, Seoul, South Korea. (page 2, line 88)

  1. line 98. Here you should specify the names of the assays and their relevant details, not in Introduction.

Thank you for your comment. According to your comment, we modified the following sentence.

The vaccinated group was evaluated using Alinity SARS-CoV-2 IgG II Quant (Abbott Laboratories, Sligo, Ireland) (RBD Ab_A), Access SARS-CoV-2 IgG (BeckmanCoulter Inc., CA, USA) (RBD Ab_B), and Elecsys anti-SARS-CoV-2 S (Roche Diagnostics GmbH, Mannheim, Germany) (RBD Ab_R) for RBD Ab assays according to the manufacturer’s instructions. (page 3, line 100)

  1. line 124. What is the substrate in this assay? Was it validated?

Thank you for your comment. We used validated commercial kit to measure nAb and modified the following sentence.

The ACE2 receptor precoated on the microplate was incubated with horseradish peroxidase(HRP)-labeled RBD, producing a strong signal. If neutralizing antibodies were present in the sample, they would attach to HRP-labeled RBD and protecting it from binding to the microplate’s ACE2 receptor. A low signal is produced by serum samples containing more nAb. (page 3, line 127)

  1. line 219. anti-SARS-Cov-2

Thank you for your comment. According to your comment, we modified the following sentence.

There are currently numerous anti-SARS-CoV-2 antibody immunoassays on the market. This can be overwhelming for laboratory directors who need to choose which test to use in routine practice. (page 9, line 226)

  1. line 222. COVID-19 is a disease, not infection.

Thank you for your comment. According to your comment, we modified the following sentence.

At the start of the pandemic, these immunoassays were primarily used to confirm SARS-CoV-2 infection and there were debates about which tests showed better performance based on N or S (RBD) proteins [25,26]. (page 9, line 228)

  1. line 227. Not true. Neutralization antibodies cant provide protection against SARS-CoV-2 without cellular immunity.

Thank you for your comment. According to your comment, we modified the following sentence.

The nAb is a key player in the humoral immune responses that defends blocking viral entry into host cells and neutralizing their biological effect in collaboration with immune cells [8]. (page 9, line 234)

  1. line 243. Why only PRT? Neutralization assay based on CPE detection is not gold standard also?

Thank you for your comment. According to your comment, we modified the following sentence.

The virus neutralization test is considered the gold-standard method to quantify nAb values against SARS-CoV-2 [10]. (page 9, line 251)

Reviewer 3 Report

Comments and Suggestions for Authors

This is well-written article.  The main benefice this study in  evaluation of three antibody binding assays,evaluation of three antibody binding assays, The authors compared the performance of the RBD Ab assay, 32 nAb assay, and IGR assay based on the vaccine types using the RBD Ab, nAb, and IGR assays. This study provides a better understanding of the assessment of humoral and cellular immune responses 34 after vaccination. Therefore this paper has interest for readers.

Author Response

I appreciate your time and effort in reviewing the manuscript, and I'm grateful for your positive response.

Round 2

Reviewer 2 Report

Comments and Suggestions for Authors

Since authors have not addressed any of my major concerns I assume that they agree that their work have big limitation and therefore we cant extrapolate any objective conslusions related to performance of examined assays.

I am therefore proposing rejection.

Author Response

We appreciate the constructive feedback provided by Reviewers and Editor on our manuscript. We designed the study to exclude subjects with an infection history to eliminate the potentiality of natural infection. This aspect has been highlighted in the revised manuscript to offer clarity on our study design and methodology. I believe these changes address the comments effectively.

Subjects were recruited and assessed for the study enrollment. Before enrollment and at every blood draw, subjects completed a questionnaire inquiring about current COVID-19 symptoms, a history of prior infection and contact with confirmed patients, adverse responses following vaccination, and underlying disease. Subjects with N Ab, RBD Ab, or nAb findings that were positive prior to vaccination were not included. Patients with clinically suspect symptoms underwent reverse transcription-polymerase chain reaction to confirm SARS-CoV-2 infection; patients exhibiting positive results were subsequently excluded. The vaccinated group without infection history and evidence included individuals who received the vector vaccine (ChAdOx1 nCoV-19, AstraZeneca, Cambridge, UK), MM vaccine (1st vaccination: ChAdOx1 nCoV-19; 2nd vaccination mRNA vaccine), or mRNA vaccine (BNT162b2, Pfizer-BioNTech, Pfizer Inc., NY, USA; mRNA-1273, Moderna, Inc., Cambridge, MA, USA). A total of 1,851 serum samples were obtained from 733 vaccinated individuals (vector, 484 subjects; MM, 145 subjects; mRNA 104 subjects). (page 2, line 88)
